# Low-Temperature Predicted Structures of Ag_2_S (Silver Sulfide)

**DOI:** 10.3390/nano13192638

**Published:** 2023-09-25

**Authors:** Stanislav I. Sadovnikov, Maksim G. Kostenko, Aleksandr I. Gusev, Aleksey V. Lukoyanov

**Affiliations:** 1Institute of Solid State Chemistry, Ural Branch of the Russian Academy of Sciences, 620990 Ekaterinburg, Russia; sadovnikov@ihim.uran.ru (S.I.S.); makskostenko@yandex.ru (M.G.K.); 2Mikheev Institute of Metal Physics, Ural Branch of the Russian Academy of Sciences, 620016 Ekaterinburg, Russia; lukoyanov@imp.uran.ru; 3Institute of Physics and Technology, Ural Federal University named after the first President of Russia B. N. Yeltsin, 620002 Ekaterinburg, Russia

**Keywords:** silver sulfide, crystal structure predictions, formation enthalpy, electronic structure, elastic constants, mechanical stability, hardness

## Abstract

Silver sulfide phases, such as body-centered cubic argentite and monoclinic acanthite, are widely known. Traditionally, acanthite is regarded as the only low-temperature phase of silver sulfide. However, the possible existence of other low-temperature phases of silver sulfide cannot be ruled out. Until now, there have been only a few suggestions about low-temperature Ag_2_S phases that differ from monoclinic acanthite. The lack of a uniform approach has hampered the prediction of such phases. In this work, the use of such an effective tool as an evolutionary algorithm for the first time made it possible to perform a broad search for the model Ag_2_S phases of silver sulfide, which are low-temperature with respect to cubic argentite. The possibility of forming Ag_2_S phases with cubic, tetragonal, orthorhombic, trigonal, monoclinic, and triclinic symmetry is considered. The calculation of the cohesion energy and the formation enthalpy show, for the first time, that the formation of low-symmetry Ag_2_S phases is energetically most favorable. The elastic stiffness constants *c_ij_* of all predicted Ag_2_S phases are computed, and their mechanical stability is determined. The densities of the electronic states of the predicted Ag_2_S phases are calculated. The prediction of low-temperature Ag_2_S structures indicates the possibility of synthesizing new silver sulfide phases with improved properties.

## 1. Introduction

Silver sulfide has three phases: low-temperature monoclinic (space group *P*2_1_/*c*) *α*-Ag_2_S (acanthite) exists at temperatures below 450 K, body-centered cubic (space group $I\;m\bar{3}m$) superionic *β*-Ag_2_S (argentite) exists in the temperature range of 452–860 K, and the high-temperature face-centered cubic *γ*-Ag_2_S phase is stable at *T* > 860 K [1,2]. When bcc *β*-Ag_2_S (argentite) is cooled below 450 K, a phase transition occurs with the formation of monoclinic *α*-Ag_2_S (acanthite) [3,4]. In cubic argentite (Ag_2±*δ*_S) S with *δ* ≈ 0.002, both a slight deficit and a slight excess of silver can be observed. Bulk monoclinic *α*-Ag_2_S (acanthite) is stoichiometric. Silver sulfide nanoparticles with a size of less than ~50 nm is nonstoichiometric. Its composition is ~Ag_1_._93_S due to vacant sites in the metal sublattice [2,5].

According to [2,6,7], the structure of *α*-Ag_2_S (acanthite) is a result of the distortion of the bcc sublattice of sulfur atoms (S) in the structure of *β*-Ag_2_S (argentite). Indeed, the unit cell of monoclinic (space group *P*2_1_/*c*) *α*-Ag_2_S (acanthite) proposed in a previous study [6] has axes that are the combinations of the axes ***a***_bcc_, ***b***_bcc_, and ***c***_bcc_ of the unit cell of bcc argentite.

The unusual physical and structural properties of the high-temperature phases, namely the enhanced ionic conductivity, the liquid-like behavior of silver sublattice [8,9], the uncertainty in the positions of silver atoms [10], and the alleged ordering, have always raised questions.

The transformation of “argentite–acanthite” is accompanied by a distortion of the bcc sublattice of S atoms to a monoclinic sublattice and displacements of sulfur and silver atoms [11]. The transformation of bcc argentite into low-temperature monoclinic acanthite can be considered a structural ordering [12,13]. It was noted in a previous study [12] devoted to the calculation and experimental determination of the Raman spectra of silver sulfide that the transformation of monoclinic silver sulfide *α*-Ag_2_S into superionic bcc argentite (*β*-Ag_2_S) is associated with disordering in the metal sublattice. As the temperature decreases below the transition temperature *T*_trans_, the S atoms, which occupied the sites of the bcc nonmetallic sublattice of argentite with equal probability, move to four sites of the monoclinic nonmetallic sublattice, leaving the remaining sites vacant (Figure 1). This relationship between the structures of *α*-Ag_2_S and *β*-Ag_2_S also raises questions regarding the existence of other ordered phases based on high-temperature *β*-Ag_2_S with the disordered silver sublattice.

This makes it possible to consider the formation of a monoclinic cell as an ordering in the bcc sublattice of sulfur atoms. Note that, in the bcc metal sublattice of argentite, upon transformation into acanthite, a peculiar ordering also occurs [14]. Indeed, silver atoms, which are randomly located with a probability of less than 0.1 in two cubic unit cells at 54 crystallographic positions of argentite, in acanthite occupy eight sites of the metal sublattice of a monoclinic unit cell with a probability of 1.

As shown earlier [13], the acanthite structure can be obtained from the argentite structure as a result of the disorder–order transition. In particular, an attempt to consider the variant of argentite ordering with the formation of a monoclinic (space group *P*2_1_) Ag_2_S phase in which the Ag atoms are in a tetrahedral environment of four S atoms was made by Kashida et al. [15]. These authors considered three hypothetical models of structural ordering for the high-temperature argentite phase, in which cooperative ionic transport is possible. These models were only used for electronic structure calculations, and an evaluation of their thermodynamic stability has not been provided. We assumed that, in addition to acanthite, other phases of Ag_2_S can form in silver sulfide, especially nanocrystalline, with a decrease in temperature.

The Open Quantum Materials Database (OQMD) [16] and Materials Project databases [17] contain many other hypothetical structures of Ag_2_S which, apparently, were simulated on the basis of structural similarity using silver chalcogenides or related systems. Both resources consider the monoclinic (space group *P*2_1_) structure proposed by Kashida et al. [15] as the best low-temperature model for silver sulfide, rather than the experimental acanthite phase. Thus, the real number of structures related exactly to Ag_2_S, as well as the anticipated sequences of phase transitions, remains unknown.

Theoretical data on the structure of several model Ag_2_S phases with triclinic, monoclinic, and orthorhombic symmetry [18,19,20,21,22,23,24,25] were calculated earlier within the framework of materials projects using the ELATE code proposed in [26]. However, the data [15,18,19,20,21,22,23,24,25] are extremely limited, often contradictory, and not entirely reliable. There is no information at all in the literature about possible highly symmetrical cubic and tetragonal Ag_2_S phases. To fill the noted gap, in this study, phases with cubic, tetragonal, orthorhombic, trigonal, monoclinic, and triclinic symmetry were considered as possible model phases of silver sulfide. The prediction of possible model phases of silver sulfide, which have a structure different from that of monoclinic acanthite, is important for obtaining the sulfide heteronanostructures of the Ag_2_S/ZnS type with a controlled band gap [27]. The band gap of the possible model phases of silver sulfide can be different from that of acanthite, which will expand the potential application of Ag_2_S/ZnS heterostructures.

Interest in the discussed low-temperature Ag_2_S polymorphs is due to the possibility of their wide application. All forms of silver sulfide (Ag_2_S) have attracted much attention [2,7,28]. Monoclinic silver sulfide (*α*-Ag_2_S) is a semiconductor at temperatures <~420–450 K, and body-centered cubic sulfide (*β*-Ag_2_S) exhibits superionic conductivity at temperatures greater than 452 K. Nanocrystalline silver sulfide is a versatile semiconductor for use in various optoelectronic devices, such as photocells, photoconductors, and infrared sensors [2,29,30]. The use of nanocrystalline silver sulfide is promising for the creation of Ag_2_S/Ag heteronanostructures intended for use in memory devices and resistance switches. Their action is based on the transformation of *α*-Ag_2_S (acanthite) into *β*-Ag_2_S (argentite) and the formation of a conducting channel between silver (Ag) and superionic *β*-Ag_2_S (argentite) [31,32,33,34]. The band gap of the predicted low-temperature Ag_2_S phases may be different from that of acanthite, which will expand the potential applications of silver sulfide.

Indeed, silver sulfide (Ag_2_S), along with lead, zinc, copper, cadmium, and mercury sulfides, is among the most in-demand semiconductors. This is an excellent material for the fabrication of heterostructures [35], which can also be used in solar cells [36], IR detectors [29,37], resistive switches, and nonvolatile memory devices [31,32,33,34]. Besides, Ag_2_S is promising for solar energy conversion into electricity [38]. The application of semiconducting nanostructured silver sulfide in biology and medicine as a biosensor is based on the quantum size effects that influence the optical properties of the material. Semiconducting Ag_2_S/noble metal heterostructures are treated as candidate materials for application in photocatalysis [39]. In particular, this is due to the narrow band gap of silver sulfide (about 0.9 eV).

The main fields of application of silver sulfide include microelectronics, biosensorics, and catalysis.

High-performance atomic resistive switches are a promising type of nonvolatile memory devices; here, read/write operations are implemented involving ion exchange [40]. In the case of cation migration, silver and silver sulfide heteronanostructures are the most widely used [31,32,33,34]. The Ag_2_S/Ag heteronanostructures combine the sulfide (semiconductor or ionic conductor, depending on the structure) and silver (electronic conductor) [41].

The design of fluorescent labels (biolabels and biomarkers) based on Ag_2_S quantum dots for applications in biology and medicine seems to be quite promising [42,43,44,45]. Nanocrystalline silver sulfide and heteronanostructures based on this substance are treated as effective materials for catalysis and photocatalysis [39,46,47]. Nanocrystalline silver sulfide has also been considered an effective thermoelectric material [48]. Silver sulfide is of interest as a thermoelectric material, owing to the reversible transition between the monoclinic semiconducting (*α*-Ag_2_S) and cubic superionic (*β*-Ag_2_S) phases. This allows the thermoelectric effect in Ag_2_S to be realized near the phase transition temperature, with silver sulfide acting as thermoelement.

Thus, silver sulfide (Ag_2_S) has a lot of possible applications. The prediction of new low-temperature phases of silver sulfide (Ag_2_S) will considerably allow for the expansion of its potential use.

The model of ordering described in previous works [2,7] implied that the initial bcc sulfur sublattice of argentite should be distorted. However, one can suggest simpler ordering models which are only based on the redistribution and “freezing” of the silver atoms, without the change in the sulfur bcc sublattice.

In addition to the conceptual structural models for ordered silver sulfide phases, we intend to use the modern technique for crystal structure prediction. In order to eliminate the inconsistencies and incompleteness in the theoretical and experimental data, we performed an extensive theoretical study of the structural properties of silver sulfide. Using a modern technique for crystal structure prediction [49], we intended to find the correct candidates for the low-temperature phases of Ag_2_S to check whether there are phases competing with acanthite at zero temperature and pressure and to compare the found structures with those given in the Materials Project database. The details of the calculations and methods we used are described in Section 2. In Section 3, we analyze the stability, electronic structure, and mechanical properties of the predicted phases. We also provide an estimation of the elastic moduli and hardness of silver sulfide since the corresponding experimental data were not available in the literature.

For the energy characterization of the model Ag_2_S phases, their cohesion energies and formation enthalpies were calculated, and the calculated elastic stiffness constants *c_ij_* were used to evaluate the mechanical stability of the suggested and predicted phases. The variation of the formation enthalpies Δ*H*_f_ of some of the predicted Ag_2_S phases with a reduction in symmetry is shown as an example in Figure 2.

## 2. Computational Simulations

When modeling the Ag_2_S phases, it was taken into account as one of the conditions that the nearest distances between Ag atoms should be greater than the doubled atomic radius of silver, equal to 0.144–0.146 nm [50], i.e., more than 0.288–0.292 nm. In other words, the Ag atoms in the model phases of silver sulfide (Ag_2_S) can be located at a distance of at least 0.288–0.292 nm from each other.

Crystal structure predictions were performed using the evolutionary algorithm (EA) USPEX (Universal Structure Predictor: Evolutionary Xtallography) [51,52,53]. The simulations were carried out for a primitive unit cell that contained one to six Ag_2_S formula units. The optimization started from a set of randomly generated structures. The number of initial random structures varied from 20 to 40, depending on the number of atoms in the cell. The structures for the next generations were produced randomly (40%) and from the previous ones (60%) via a set of flexible physically motivated variation operators (60%) [51,52,53].

The total energies of the predicted model structures were calculated according to density functional theory [54] using the Perdew–Burke–Ernzenhof version of the generalized gradient approximation (GGA) [55] and the projector-augmented wave method [56] as implemented in the VASP (Vienna ab initio simulation package) code [57,58]. To find equilibrium unit cell parameters and atomic positions, we performed stepwise relaxation [49] using VASP-implemented conjugate-gradient and quasi-Newton RMM-DIIS algorithms [59]. The energies of the initial USPEX-generated structures were computed using a kinetic energy cutoff of 259 eV and uniform *k*-point meshes with a reciprocal space resolution of 2*π* × 0.12 Å^−1^. During the following relaxation steps, these values were gradually improved to 388 eV and 2*π* × 0.05 Å^−1^, respectively. The structures with minimal energy generated using USPEX were finally treated at a denser *k*-point grid with a resolution of 2*π* × 0.03 Å^−1^.

In order to estimate the thermodynamic stability of the silver sulfide (Ag_2_S) model phases with different crystal structures, we calculated their enthalpies of formation at *T* = 0 K and *P* = 0 GPa (Table 1). Metallic silver (Ag) (space group No.225-$F\;m\bar{3}m$) and orthorhombic sulfur (S) (space group No.70-*Fddd*) were considered as initial reactants. The formation enthalpies Δ*H*_f_ of the predicted Ag_2_S phases according to the reaction
2Ag + S = Ag_2_S(1)
were determined, taking into account the real number of atoms in the unit cells of Ag, S, and Ag_2_S. The unit cell of metallic silver contains four atoms of Ag, the unit cell of sulfur (S) contains 128 atoms of S, and the predicted Ag_2_S phase contains *Z* formula units. The number of Ag_2_S formula units coincides with the number of S atoms in the predicted phase. Taking this into account, the formation enthalpy Δ*H*_f_ of the predicted Ag_2_S phase according to reaction (1) was found using the formula
(2)$$\Delta H_{f}=[E_{\mathrm{phase}}-(N_{\mathrm{Ag}}E_{\mathrm{Ag}-\mathrm{cond}}/N_{\mathrm{Ag}-\mathrm{cond}})-(N_{S}E_{S-\mathrm{cond}}/N_{S-\mathrm{cond}})]/Z,$$
where *E* is the energies of the predicted Ag_2_S phase, condensed metallic silver (Ag) with a cubic (space group $F\;m\bar{3}m$) structure, and the condensed sulfur (S) phase with an orthorhombic (space group *Fddd*) structure obtained via the DFT calculations. In Formula (2), *N*_Ag_ and *N*_S_ are the numbers of these atoms in the unit cell of the predicted Ag_2_S phase; *Z* is the number of formula units in the predicted Ag_2_S phase. The values of *N*_Ag-cond_, *N*_s-cond_, *E*_Ag-cond_, and *E*_s-cond_ are given in Appendix A. The division by *Z* normalizes the Δ*H*_f_ values to an Ag_2_S formula unit (i.e., to an S atom).

The cohesion energy *E*_coh_ is an alternative parameter to characterize the energy advantage of the predicted phase. In contrast to the formation enthalpy, *E*_coh_ shows the energy gain in the formation of a substance from individual atoms:(3)$$E_{\mathrm{coh}}=(E_{\mathrm{phase}}-N_{\mathrm{Ag}}E_{\mathrm{Ag}-\mathrm{at}}-N_{S}E_{S-\mathrm{at}})/Z,$$
where *E*_phase_ is the energy of the ground state of the predicted Ag_2_S phase according to DFT calculations, and *E*_Ag-at_ and *E*_S-at_ are the energies of individual Ag and S atoms calculated as the energies of simple cubic structures, where atoms are separated by a vacuum to exclude their interaction. The values *E*_Ag-at_ and *E*_S-at_ of individual Ag and S atoms are given in Appendix A.

For example, the cubic (space group $Pn\bar{3}m$) Ag_2_S structure is characterized by the following quantities: *E*_phase_ = −19.498 eV, *N*_Ag_ = 4, *N*_S_ = 2, and *Z* = 2 (see Table 1), and *E*_Ag-cond_ = −10.863 eV, *N*_Ag-cond_ = 4, *E*_S-cond_ = −528.168 eV, *N*_S-cond_ = 128, *E*_Ag-at_ = −0.198 eV, and *E*_S-at_ = −1.081 eV (see Appendix A). According to Equation (2), the formation enthalpy Δ*H*_f_ of the cubic (space group $Pn\bar{3}m$) Ag_2_S structure for these quantities is
Δ*H*_f_ = [−19.498 − (−10.863·4/4) − (−528.168·2/128)]/4 = [−19.498 + 10.863 + 8.252]/4 = −0.382/2 = −0.191 eV. 
According to Equation (3), the cohesion energy *E*_coh_ of the cubic (space group $Pn\bar{3}m$) Ag_2_S structure for these quantities is
*E*_coh_ = [−19.498 − (−0.198·4) − (−1.081·2)]/2 = [−19.498 + 0.792 + 2.162]/2 = −16.544/2 = −8.272 eV. 

It is these values of Δ*H*_f_ and *E*_coh_ that are given in Table 1 for the cubic (space group $Pn\bar{3}m$) Ag_2_S structure.

The structures predicted using the USPEX code were compared with the experimental acanthite phase, as well as the ordered models. To ensure the accuracy of this comparison, we optimized the unit cell parameters and positions of atoms using the identical settings of the DFT calculations of their energies. The structural optimization stopped if the forces acting on the atoms were less than 10^−3^ eV·Å^−1^. The forces were calculated according to the Hellmann–Feynmann theorem.

The DOS diagrams were computed using the tetrahedron method with Blöchl corrections [60]. The band structure curves were built according to the *k*-points trajectories suggested using the SeeK-path tool [61]. The elastic properties and mechanical stability of the predicted model Ag_2_S structures were estimated based on elastic tensor coefficients *c_ij_* computed using the finite difference method as implemented in the VASP code [56,57,58]. The mechanical stability of the model structures was determined using the necessary and sufficient criteria given in previous works [62,63].

## 3. Results and Discussion

### 3.1. Structures and Energies of the Predicted Ag_2_S (Silver Sulfide) Phases

The literature’s theoretical data on the structure and lattice parameters of some model Ag_2_S phases with triclinic, monoclinic, and orthorhombic symmetry are presented in the form of a database on the websites of the Materials Project, such as mp-32284 [18], mp-556225 [19], mp-1095694 [20], mp-32669 [21], mp-36216 [22], mp-31053 [23], mp-32791 [24], and mp-610517 [25], and others. The crystallographic data of these silver sulfide phases were calculated ab initio using the ELATE code, described in general form in a previous study [26].

The main energy characteristics of the model Ag_2_S phases predicted using the USPEX code are given in Table 1.

The crystallographic data for the considered cubic, tetragonal, and trigonal phases can be found in Appendix A, respectively (see Appendix A). The crystallographic data for the orthorhombic and monoclinic (space group *P*2_1_/*c*) *α*-Ag_2_S (acanthite) structure before and after relaxation, the predicted monoclinic (space group *P*2_1_/*c*) *α*-Ag_2_S structure, and the predicted triclinic (space group *P*1) Ag_2_S structure are given in Table 2, Table 3, Table 4 and Table 5.

The unit cells of the model cubic, tetragonal, and trigonal Ag_2_S phases, which can presumably form as alternative low-temperature phases along with the monoclinic acanthite (*α*-Ag_2_S), are shown in Figure 3a–c, respectively. All the model crystal structures were visualized using the VESTA software [64]. The translation vectors and atomic coordinates in the unit cells of these model Ag_2_S phases are given in Appendix A, respectively (see Appendix A).

The formation enthalpies Δ*H*_f_ of the model cubic phases of Ag_2_S with space groups $Pn\bar{3}m$ and $Fd\bar{3}m$ (see Figure 3a) are −0.191 and +3.573 eV/form.unit (see Table 1). It is clear that the formation of the model cubic (space group $Fd\bar{3}m$) Ag_2_S phase is thermodynamically impossible, while the cubic (space group $Pn\bar{3}m$) Ag_2_S structure is quite favorable in terms of the enthalpy of formation. This cubic phase is markedly more favorable than the unrelaxed acanthite structure, but it is somewhat inferior to the relaxed acanthite structure (see Table 1).

The model tetragonal structures of Ag_2_S with space groups $P\bar{4}c2$ and *P*4/*mmm* calculated using the USPEX code are shown in Figure 3b. The translation vectors and atomic coordinates in the unit cells of these tetragonal Ag_2_S phases are given in Table 3. The suggested tetragonal structures with space groups *P*4/*mmm* and $P\bar{4}c2$ have positive formation enthalpies of +1.313 and +0.394 eV/form.unit, respectively (see Table 1). Therefore, the formation of such tetragonal structures of silver sulfide is energetically impossible.

A similar conclusion should be drawn for the trigonal (space groups $R\bar{3}$ and $R\bar{3}m$) structures with the formation enthalpies of +0.042 and +0.041 eV/form.unit, respectively (see Table 1). The unit cells of these trigonal structures of Ag_2_S are shown in Figure 3c, and the translation vectors and atomic coordinates for these trigonal Ag_2_S phases are given in Appendix A.

Figure 4 shows the model orthorhombic Ag_2_S structures with space groups *Cmce* and *Cmcm*. The translation vectors and atomic coordinates in the unit cells of these orthorhombic Ag_2_S phases are given in Table 2. The calculated formation enthalpies Δ*H*_f_ of these structures are −0.219 and −0.199 eV/form.unit, respectively (see Table 1), so their formation is quite possible.

Using the USPEX code, monoclinic (space group *P*2_1_/*c*) models of the α-Ag_2_S (acanthite) structure were calculated before and after relaxation (Figure 5). The initial relaxation was carried out in terms of energy, and then the structure was recalculated with more accurate convergence criteria. Translation vectors and atomic coordinates in the unit cells of the unrelaxed and relaxed monoclinic (space group *P*2_1_/*c*) phases of silver sulfide with the *α*-Ag_2_S (acanthite) structure are given in Table 3.

The first point we should pay attention to is a rather uncommon behavior of the acanthite structure. Its structure, obtained using an X-ray diffraction experiment (see Figure 5a), differs significantly from the structure corresponding to its DFT energy minimum (see Figure 5b). Although, during structural relaxation, the symmetry was fixed, and both variants of acanthite are of the same space group, *P*2_1_/*c*, they are somewhat different (see Figure 5). The result of structural optimization for acanthite (Figure 5b) does not depend on whether energies or forces were used as a minimization criterion.

The relaxed monoclinic (space group *P*2_1_/*c*) model of acanthite (*α*-Ag_2_S) turned out to be the best in terms of the formation enthalpy. The formation enthalpy Δ*H*_f_ of the unrelaxed acanthite structure is −0.033 eV/form.unit, while the enthalpy Δ*H*_f_ of formation of the relaxed acanthite structure is much lower and amounts to −0.199 eV/form.unit (see Table 1).

We calculated the X-ray diffraction (XRD) patterns of unrelaxed and relaxed monoclinic (space group *P*2_1_/*c*) *α*-Ag_2_S (acanthite) (Figure 6). The calculations were carried out under *CuKα*_1,2_- radiation with a Δ(2*θ*) = 0.01° step. Figure 6 shows sections of the X-ray diffraction patterns in the range of angles 2*θ* = 19–61°, where the most characteristic and intense diffraction reflections of these acanthite phases are present. The calculated XRD pattern of unrelaxed acanthite coincides with high accuracy with the experimental XRD patterns of coarse-grained monoclinic (space group *P*2_1_/*c*) *α*-Ag_2_S (acanthite) [2,65]. After the relaxation of acanthite, the unit cell parameters and, accordingly, the coordinates of the Ag and S atoms somewhat changed (Table 6). As a result of atomic displacements, a more ordered arrangement of atoms, especially Ag atoms, was observed in the relaxed phase compared to unrelaxed acanthite (see Figure 5). In particular, Ag atoms are predominantly located in atomic planes perpendicular to the a and c axes of the unit cell. This arrangement of atoms led to a noticeable decrease in the number of diffraction reflections in the XRD pattern of relaxed acanthite, although some of the reflections coincide in position with the reflections of the unrelaxed phase (indices of coinciding reflections are shown in Figure 5). Relaxed acanthite clearly exhibits interplanar distances of ~0.390, ~0.335, ~0.277, ~0.230, and ~0.210 nm, which are also characteristic of the unrelaxed phase. Note that, in numerous experimental studies of silver sulfide summarized in [2,7], only unrelaxed acanthite was observed as a low-temperature phase, regardless of the method and conditions of synthesis.

The calculation of other model Ag_2_S structures showed that the monoclinic structure of acanthite (*α*-Ag_2_S) described in the literature [6,65] is not the only possible and most energetically favorable low-temperature phase of silver sulfide. Because of calculations, it was possible to find a monoclinic (space group *P*2_1_/*c*) Ag_2_S phase (Figure 7, left) with lower energy in the ground state compared to relaxed acanthite. The translation vectors and atomic coordinates in the unit cells of this monoclinic (space group *P*2_1_/*c*) phase of silver sulfide are given in Table 4. The formation enthalpy Δ*H*_f_ of this monoclinic (space group *P*2_1_/*c*) Ag_2_S phase is −0.219 eV/form.unit (see Table 1), i.e., one 0.02 eV/form.unit less than the enthalpy of relaxed acanthite.

As a result of the performed simulation of other possible Ag_2_S structures, it was possible to establish that the monoclinic phases of silver sulfide (Ag_2_S) are not the most energetically favorable low-temperature phases. The calculations allowed us to find the triclinic (space group *P*1) Ag_2_S phase (Figure 7, right). This Ag_2_S phase has the lowest cohesion energy *E*_coh_ = −8.304 eV/form.unit and formation enthalpy Δ*H*_f_ = −0.223 eV/form.unit (see Table 1) in the ground state at *T* = 0 K and *P* = 0 Pa.

Translation vectors and atomic coordinates in the unit cell of the triclinic (space group *P*1) Ag_2_S phase are given in Table 5.

As for the USPEX predicted structures, ten of them have formation enthalpies significantly lower compared to unrelaxed acanthite and can be considered alternative low-temperature Ag_2_S modifications of silver sulfide (see Table 1). The triclinic (space group *P*1) structure (see Figure 7, right) has the lowest formation enthalpy (−0.223 eV/form.unit) of all the considered modifications. The other nine phases, despite the variety in their symmetries, have very similar structures. Their different symmetry is a result of small variations in angles and/or bond lengths. The monoclinic (space group *P*2_1_) Ag_2_S phase proposed by Kashida et al. [15] also has a similar structure. The formation enthalpy of this monoclinic (space group *P*2_1_) Ag_2_S phase is −0.208 eV/form.unit (see Table 1), which is slightly lower than the Δ*H*_f_ of relaxed acanthite. The lowest formation enthalpy of −0.219 eV/form.unit for these almost identical structures is reached for the orthorhombic (space group *Cmce*) Ag_2_S phase (see Figure 4). The other examined structures of Ag_2_S with space groups *Cmcm*, *P*2_1_/*m*, and *Cmc*2_1_ from the Materials Project database are merely variants of relaxed acanthite (see Figure 5b) with formation enthalpies ranging from −0.1993 to −0.1990 eV/form.unit.

We assessed the influence of residual forces on the standard deviation of the formation enthalpies of the considered Ag_2_S phases. In particular, when assessing the impact of the residual forces on the unrelaxed acanthite structure, we found that the absolute deviations in formation enthalpy did not exceed 0.00034 eV/(Ag_2_S form.unit) or 0.00085 eV·Å^−1^·atom^−1^ when changing the force threshold from 0.002 eV·Å^−1^ to 0.001 eV·Å^−1^. The relative deviation of the formation enthalpy does not exceed 0.1%. For smaller values of the limiting parameter, it was impossible to complete the relaxation of the structure due to computational limitations. The other considered phases were more susceptible to relaxation, so deviations in their energies are very likely to be within this 0.1% range. This amount of uncertainty is enough to rank the Ag_2_S structures by their stability. For instance, the difference in the formation enthalpies, when taking into account their uncertainties, for the two best structures (triclinic and orthorhombic with space groups *P*1 and *Cmce*) is about 2%. Thus, the deviations in the formation enthalpies of structures with different symmetries are very small, which makes it possible to distinguish between these structures.

The Ag-S bond lengths for all the predicted Ag_2_S structures are given in Appendix A. The Ag-S bond length is 0.232–0.252 nm, depending on the symmetry of the predicted Ag_2_S structure.

In high-temperature body-centered cubic *β*-Ag_2_S (argentite) at a temperature close to 453 K, the length of the bonds Ag1-S and Ag2-S are 0.24307 and 0.25691nm, respectively. The coordination numbers of the Ag1 (*b* positions) and Ag2 (*j* positions) atoms are 6 and 1, respectively. In the crystal lattice of *α*-Ag_2_S acanthite, the length of the bonds Ag1-S and Ag2-S are 0.25113 and 0.25475 nm, respectively (see Appendix A). The coordination numbers of the Ag1 and Ag2 atoms are the same and equal to 4. Thus, when taking into account the temperature change in the lattice parameters of silver sulfide during the transition from argentite to acanthite, the Ag1-S and Ag2-S bond lengths increased, and the coordination numbers of the Ag atoms became the same. The coordination numbers of the silver atoms in the predicted Ag_2_S structures are also less than 6. The smallest coordination number of Ag atoms, equal to 1, was observed for Ag-S1 bonds with a length of 0.23883 nm in the cubic (space group $Fd\bar{3}m$) Ag_2_S phase. Coordination numbers of Ag atoms equal to 2 were observed for Ag-S bonds with lengths of 0.23556 nm and 0.24306 nm in the cubic (space group $Pn\bar{3}m$) and orthorhombic (space group *Cmce*) Ag_2_S phases, as well as for all Ag-S bonds in a predicted triclinic (space group *P*1) Ag_2_S phase. The coordination number of Ag atoms in the Ag-S bonds of tetragonal (space group $P\bar{4}c2$) and trigonal (space groups $R\bar{3}$
*and*
$R\bar{3}m$) silver sulfides is 3. A coordination number of Ag atoms equal to 3 was also observed in relaxed monoclinic (space group *P*2_1_/*c*) acanthite. Ag atoms have a fourfold coordination environment by S atoms in unrelaxed monoclinic (space group *P*2_1_/*c*) acanthite and tetragonal (space group *P*4/*mmm*) silver sulfide.

In the structures with the lowest formation enthalpies (triclinic and orthorhombic with space groups *P*1 and *Cmce*), the Ag-S bond lengths range from 0.2431 to 0.2496 nm. The Ag-S bond lengths of the relaxed structure of acanthite are somewhat outside these limits and demonstrate an increasing disproportion. One of the bonds has a length of 0.2524 nm, while another 0.2403 nm. This disproportion is considerably larger in the other proposed ordered structures. The biggest difference in bond lengths is reached for the tetragonal (space group $P\bar{4}c2$) and one of the cubic (space group $Fd\bar{3}m$) models. Therefore, from the chemical point of view, the formation of such structures seems unreasonable. The cubic (space group $Fd\bar{3}m$) structure after relaxation contains chemically isolated sulfur atoms. The nearest distance to the silver atom is 0.6545 nm. Presumably, this peculiarity makes the cubic (space group $Fd\bar{3}m$) structure the most unfavorable of all the considered structural models of silver sulfide. Another suggested cubic (space group $Pn\bar{3}m$) structure has no disproportion in bond lengths. Its nearest Ag-S distance (0.2356 nm) is slightly less than that of the best structures. Nevertheless, the formation energy of the cubic (space group $Pn\bar{3}m$) ordered structure is close to that of acanthite. Therefore, of all the suggested cubic, tetragonal, and trigonal models, only the cubic (space group $Pn\bar{3}m$) structure can be considered a candidate for a new phase of silver sulfide.

As a result of the simulations and analysis, we should conclude that acanthite is not the most energetically favorable low-temperature variant in the crystal structure of silver sulfide. The predictions based on the evolutionary algorithm allowed us to find two new Ag_2_S phases (see Figure 7) with lower formation energies.

### 3.2. Symmetry Analysis of the Predicted Structures

Crystallographic point and space groups are comprehensively described in monographs [66,67,68,69]. In particular, all point groups and all their symmetry elements are listed in a monograph [69] in accordance with the notation adopted in [67]: forty-eight elements of the full cubic symmetry group $m\bar{3}m$ (*O_h_*) are sequentially denoted from *h*_1_ to *h*_48_.

The point group $m\bar{3}m$ (*O_h_*) of bcc argentite (*β*-Ag_2_S) includes all 48 symmetry elements *h*_1_-*h*_48_ of the cubic group [66], i.e., *N_h_*_-arg_ = 48. The rotational reduction in symmetry *N*_rot_ is equal to the ratio of the number of symmetry elements of the argentite point group to the number *N_h_*_-phase_ of symmetry elements included in the point group of the Ag_2_S model structure under consideration, i.e., *N*_rot_ = *N_h_*_-arg_/*N_h_*_-phase_ = 48/*N_h_*_-phase_. The reduction in translational symmetry is equal to the relative change in unit cell volume during the transformation of argentite into the Ag_2_S model structure. During the transformation of one phase of silver sulfide (in our case, argentite) into another Ag_2_S phase, the decrease in the translational symmetry *N_tr_* can also be estimated as the ratio of the *Z*_phase_ value (the number of Ag_2_S formula units in the discussed model phase) to the number of Ag_2_S formula units in argentite (Z_arg_ = 2), i.e., *N*_tr_ = *Z*_phase_/*Z*_arg_. However, the decrease in translational symmetry is more accurately estimated from the relative change in unit cell volumes during the considered transformation. The total symmetry reduction *N*_tot_ = *N*_rot_ × *N*_tr_ is the product of the rotational symmetry reduction and the translational symmetry reduction. For example, in the transition of argentite to monoclinic *α*-Ag_2_S (acanthite), which contains *Z*_acanth_ = 4 Ag_2_S formula units and whose point group 2/*m* (*C*2*_h_*) includes four symmetry elements, *h*_1_, *h*_4_, *h*_25_, and *h*_28_ (*N_h_*_-acanth_ = 4), the total symmetry reduction *N*_tot_ = 24.

The decrease in the symmetry *N*_tot_ of the predicted model Ag_2_S phases with respect to argentite is given in Table 1. When using the example of silver sulfide structures with the same monoclinic (space group *P*2_1_/*c*) symmetry, it is clearly seen that the largest decrease in symmetry leads to the formation of phases with the lowest formation enthalpy (see Table 1).

In general, calculations of the model structures of silver sulfide (Ag_2_S) using the USPEX code [51,52,53] showed that a reduction in the symmetry of Ag_2_S phases from cubic, tetragonal, and trigonal to orthorhombic, monoclinic, and especially triclinic crystallographic systems is accompanied by a decrease in their formation enthalpy Δ*H*_f_ and the emergence of the most energetically favorable structures. This was not previously known. Indeed, the transformation of high-temperature cubic (space group $I\;m\bar{3}m$) *β*-Ag_2_S (argentite) into any model Ag_2_S structure will occur with a decrease in symmetry.

### 3.3. Electronic Structure

The density of electron states (DOS) calculations for the predicted Ag_2_S phases were performed per electron density functional theory (DFT) [54] using the tetrahedron method with Blöchl corrections [60]. The exchange-correlation potential was described using the PBE [55] version of the generalized gradient approximation (GGA). The band structure was built along the trajectories of *k*-points determined using the SeeK-path service [61]. The kinetic energy of plane waves did not exceed 388 eV.

All the considered low-temperature Ag_2_S phases are very similar in their electronic structures. Their band structures exhibit a band gap with a width of ~0.6 to ~1.5 eV, which indicates the semiconductor properties of the predicted Ag_2_S phases. For relaxed Ag_2_S structures, the band gap *E*_g_ varies from 1.02 to 1.16 eV and is close to the experimentally measured values of 0.9–1.1 eV [2,7] for nanostructured silver sulfide.

The band structure of unrelaxed and relaxed monoclinic (space group *P*2_1_/*c*) *α*-Ag_2_S (acanthite) is shown in Figure 8. It is clearly seen that unrelaxed acanthite is a direct-gap semiconductor. Because of relaxation, the band structure changed somewhat, since the electron located near the bottom of the conduction band acquired a momentum that differed from the momentum of the electron located near the maximum of the valence band. Thus, relaxed acanthite becomes an indirect-gap semiconductor.

The densities of states of unrelaxed and relaxed monoclinic (space group *P*2_1_/*c*) *α*-Ag_2_S (acanthite) are shown in Figure 9. The structure of unrelaxed acanthite, formed from X-ray diffraction data [46], has a much smaller band gap of 0.72 eV compared to *E*_g_ = 1.02 eV for relaxed acanthite (*α*-Ag_2_S). The states near the Fermi level are formed by Ag 4*d* orbitals, and the S 3*p* states are most pronounced in the energy ranges below ~0.5–2.0 and ~4.5–6.0 eV relative to the Fermi level. The edge of the valence band of monoclinic (space group *P*2_1_/*c*) acanthite near the Fermi level has a slope and resembles experimental optical spectra [70,71,72]. The width of the valence band of acanthite is approximately equal to 4–6 eV, which coincides with the experimental data on the photoelectron spectroscopy of silver sulfide [13,73].

The predicted cubic (space group $Pn\bar{3}m$) Ag_2_S structure was proposed as a derivative of the bcc (space group $I\;m\bar{3}m$) structure of *β*-Ag_2_S (argentite). The formation enthalpy Δ*H*_f_ = −0.191 eV/(Ag_2_S form.unit) of this structure is one of the lowest among the predicted silver sulfide phases. Nevertheless, its electronic structure (see Figure 9c) differs markedly from that of other low-temperature Ag_2_S phases. The valence band of the cubic (space group $Pn\bar{3}m$) Ag_2_S structure is split, and the band gap is 1.54 eV, i.e., about 50% more than that of other predicted Ag_2_S structures.

### 3.4. Elastic Properties and Hardness

The elastic properties of the model cubic, tetragonal, trigonal, orthorhombic, monoclinic, and triclinic Ag_2_S structures were estimated by calculating the coefficients of elasticity tensors. The found matrices (***C***) of elastic stiffness constants *c_ij_* (S6)–(S17) of the predicted model phases of Ag_2_S (silver sulfide) are given in the Appendix A.

The analytical relationships between compliance coefficients *s_ij_* and stiffness coefficients *c_ij_* for low-symmetry (trigonal, monoclinic, and triclinic) crystals are very cumbersome and complex. For this reason, in general cases, the elastic compliance constants *s_ij_* are found by calculating the matrix (***S***) of the compliance constants, which is inverse to the matrix (***C***), i.e., (***S***) = (***C***)^−1^. We found the matrices (***S***) of the predicted Ag_2_S structures by calculating the inverse matrices using the software [74] for the calculation of inverse matrices.

The calculated elastic constants *c_ij_* (see matrices (S6)–(S17)) were used to evaluate the mechanical stability of the predicted model Ag_2_S phases using the Born criteria [75,76] and the necessary and sufficient conditions for the elastic stability of various crystal systems described in studies [62,63]. The calculated elastic stiffness constants *c_ij_* and compliance constants *s_ij_* were also used to estimate the isotropic elastic moduli of some predicted Ag_2_S phases of silver sulfide.

A necessary but not sufficient condition for the mechanical stability of a crystal of any symmetry is the positiveness of all diagonal elements of the matrix of elastic stiffness constants, i.e., *c_ii_* > 0 (*i* = 1–6).

The stability criteria for cubic crystals have the form *c_ii_* > 0, *c*_11_ > *c*_12_, *c*_44_ > 0, and *c*_11_ + 2*c*_12_ > 0 [62,63,75]. These criteria are met for the cubic (space group $Fd\bar{3}m$ and $Pn\bar{3}m$) Ag_2_S model phases. Thus, the cubic (space group $Pn\bar{3}m$) Ag_2_S phase is mechanically stable. However, the positive formation enthalpy of a cubic phase with a space group $Fd\bar{3}m$ excludes the existence of such a phase.

The mechanical stability conditions for tetragonal crystals have the form *c*_11_ > |*c*_12_|, *c*_44_ > 0, *c*_66_ > 0, and (*c*_11_ + *c*_12_)*c*_33_ > $2c_{13}^{2}$ [43,44]. The elastic stiffness constants of the model tetragonal (space group *P*4/*mmm*) Ag_2_S phase do not satisfy these conditions since *c*_44_ < 0 and *c*_55_ < 0 (see matrix (S9), Appendix A), so this tetragonal (space group *P*4/*mmm*)) Ag_2_S phase is mechanically unstable. The elastic stiffness constants of the tetragonal (space group $P\bar{4}c2$) Ag_2_S phase also do not satisfy these conditions since *c*_11_ < *c*_12_ and (*c*_11_ + *c*_12_)*c*_33_ < $2c_{13}^{2}$ (see matrix (S8), Appendix A), so this tetragonal (space group $P\bar{4}c2$) Ag_2_S phase is also mechanically unstable.

The mechanical stability conditions for trigonal (rhombohedral) crystals have the form *c*_11_ > |*c*_12_|, *c*_44_ > 0, (*c*_11_ + *c*_12_)*c*_33_ > $2c_{13}^{2}$, and (*c*_11_ − *c*_12_)*c*_44_ > $2c_{14}^{2}$ + $2c_{15}^{2}$. The absence in the elastic matrices of the trigonal (space groups $R\bar{3}$ and $R\bar{3}m$) Ag_2_S phases of elastic stiffness constants equal to 0, and the presence of negative elastic stiffness constants where there should be zero constants (see matrices (S10) and (S11), Appendix A), means that the mechanical stability conditions are not satisfied for these phases. Thus, the model trigonal (space groups $R\bar{3}$ and $R\bar{3}m$) Ag_2_S phases are mechanically unstable.

Necessary and sufficient conditions for the mechanical stability of orthorhombic crystals have the form *c_ii_* > 0 (*i* = 1–6), *c*_11_*c*_22_ > $2c_{12}^{2}$, and *c*_11_*c*_22_*c*_33_ + 2*c*_12_*c*_13_*c*_23_ − $c_{11}c_{23}^{2}$ − $c_{22}c_{13}^{2}$ − $c_{33}c_{12}^{2}$ > 0 [62,63] and are satisfied for the orthorhombic (space groups *Cmcm* and *Cmce*) Ag_2_S phases.

For the model monoclinic (space group *P*2_1_) Ag_2_S phase, and also for unrelaxed and relaxed monoclinic (space group *P*2_1_/*c*) *α*-Ag_2_S (acanthite), the elastic stiffness constants *c_ii_* > 0, *c*_22_ + *c*_33_ > 2*c*_23_, *c*_11_ + *c*_22_ + *c*_33_ > 2(*c*_12_ + *c*_13_ + *c*_23_), *c*_33_*c*_55_ > $c_{35}^{2}$, and *c*_44_*c*_66_ > $c_{46}^{2}$ (see matrices (S14)–(S16) in Appendix A) satisfy the necessary and sufficient conditions for mechanical stability. Thus, the monoclinic (space groups *P*2_1_ and *P*2_1_/*c*) Ag_2_S phases with such elastic stiffness constants are mechanically stable.

For the predicted triclinic (space group *P*1) Ag_2_S phase, the elastic stiffness constants *c_ii_* > 0 (see matrix (S17), Appendix A). With this in mind, we can assume that the predicted triclinic (pace group *P*1) Ag_2_S phase satisfies the conditions of mechanical stability and is mechanically stable.

Thus, the mechanical stability of all the predicted Ag_2_S phases was determined for the first time. Previously, there was no information on the mechanical stability of low-temperature Ag_2_S phases.

The hardness of silver sulfide on the Moh’s scale is 2.0–2.5. There is no information on the Vickers hardness (*H*_V_) of silver sulfide (Ag_2_S) in the accessible literature. The availability of calculated data on the constants of the elastic stiffness *c_ij_* and elastic compliance *s_ij_* of the considered model Ag_2_S phases makes it possible to find the isotropic elastic moduli and hardness of these phases. We evaluated the isotropic elastic moduli *E*_H_, *G*_H_, and *B*_H_ and the hardness (*H*_V_) of several predicted Ag_2_S phases that have the lowest formation enthalpies.

Pugh [77] suggested the use of the bulk modulus (*B*) to shear modulus (*G*) ratio as a criterion for the brittle and ductile behavior of polycrystalline metals. The isotropic elastic moduli of polycrystalline materials can be averaged based on the upper (*B*_V_) and lower (*B*_R_) limit values of the bulk modulus, and the upper (*G*_V_) and lower (*G*_R_) values of the shear modulus. The values of *B*_V_, *B*_R_, *B*_H_, *G*_V_, *G*_R_, and *G*_H_ for crystals with any symmetry are calculated via the Voigt–Reuss–Hill method [78] (see Appendix A) using elastic stiffness constants *c_ij_* and compliance constants *s_ij_*. The Poisson’s ratio *μ* was calculated as *μ* = (3*B*_H_ − 2*G*_H_)/[2(3*B*_H_ + *G*_H_)]. The calculated isotropic elastic moduli for *G*_H_, *B*_H_, *E*_H_ of several polycrystalline silver sulfides (Ag_2_S) with the lowest formation enthalpies are given in Table 6.

The isotropic elastic moduli *G*_H_ = 9.8–10.1 and *B*_H_ = 28.3–31.1 GPa of unrelaxed and relaxed monoclinic (space group *P*2_1_/*c*) *α*-Ag2S (acanthite) (see Table 6) are in good agreement with the values *G*_H_ = 8 and *B*_H_ = 26 GPa of the monoclinic (space group *P*2_1_/*c*) phase of silver sulfide given in the Materials Project’s mp-610517 [25]. The *B*_H_ modulus of acanthite calculated by us is 28–31 GPa (see Table 6) and is quite close to the experimental estimate of *B* = 33.7 GPa [79] at 0 K, determined from data on the temperature dependences of the heat capacity and thermal expansion coefficient. The calculated isotropic moduli *G*_H_ = 10.2 and *B*_H_ = 28.4 GPa of the predicted orthorhombic (space group *Cmcm*) Ag_2_S (silver sulfide) are close to the values of *G*_H_ and *B*_H_ found for the same orthorhombic sulfide (Ag_2_S) in [22] and equal to 9 and 24 GPa, respectively.

Currently, the inverse Pugh’s ratio, *k* = *G*/*B*, is used to estimate the theoretical Vickers hardness (*H*_V_) of solids. Chen et al. [80] proposed describing the Vickers hardness (*H*_V_) of solids as a function of the inverse Pugh’s ratio *k* and shear modulus *G* and bulk modulus *B*:*H*_V_ = 2(*k*^2^*G*)^0^*^.^*^585^ − 3.(4)
However, later, Tian et al. [81] noted that there are no physical grounds for using the free negative term −3 in Formula (4) since this leads to negative hardness values for some ionic and other crystals. Therefore, Tian et al. [81] proposed another formula for estimating the hardness *H*_V_:*H*_V_ = 0.92*k*^1.37^*G*^0*.*708^.(5)

In Formulas (4) and (5), the isotropic elastic moduli *B*_H_ and *G*_H_ of polycrystalline materials are used to evaluate hardness.

Later, Mazhnik and Oganov [82] suggested using Young’s modulus *E* instead of *B*_H_ and *G*_H_, and they proposed using the empirical formula
*H*_V_ = *γ*_0_*χ*(*μ*)*E*,(6)
where *γ*_0_ = 0.096 is a dimensionless constant independent of the material, and *χ*(*μ*) is a function that depends on the value of the Poisson’s ratio *μ* and has the form
(7)$$\chi (\mu )=\frac{1-8.5\mu +19.5\mu^{2}}{1-7.5\mu +12.2\mu^{2}+19.6\mu^{3}}.$$

The found isotropic elastic moduli *G*_H_, *B*_H_, and *E*_H_ of polycrystalline silver sulfides (Ag_2_S), the predicted structures of which have the lowest enthalpies of formation, are presented in Table 6. The use of Formula (4) led to negative hardness values for some silver sulfides, so we calculated the hardness using Equations (5) and (6). The hardness *H*_V_ of polycrystalline silver sulfides (Ag_2_S), calculated using Equation (5), is from 1.0 to 3.2 GPa (Table 6). The use of Equation (6) resulted in slightly lower values, ranging from 0.9 to 1.5 GPa (see Table 6). The estimated hardness of silver sulfides is higher than the hardness of metallic silver and such metals as Fe, Ni, and Ti, and it is close to the hardness of calcium carbonate and fluoride (~1 and ~2 GPa, respectively).

Thus, for the first time, the isotropic elastic moduli of the predicted polycrystalline Ag_2_S phases were determined, and the hardness *H*_V_ of the predicted Ag_2_S phases with the lowest formation enthalpies was estimated.

## 4. Conclusions

In this work, we considered alternative structural models for low-temperature modifications of silver sulfide (Ag_2_S) in addition to the known monoclinic structure of acanthite. The proposed structures, like acanthite itself, can be represented as the results of the ordering of the high-temperature cubic argentite phase. The possibility of the formation of cubic, tetragonal, orthorhombic, trigonal, monoclinic, and triclinic Ag_2_S phases was considered. The calculation of the cohesion energy and the formation enthalpy showed that the formation of low-symmetry Ag_2_S phases is energetically the most favorable. The most favorable phase with the lowest formation enthalpy has triclinic (space group *P*1) symmetry. The elastic stiffness constants *c_ij_* of all the predicted Ag_2_S phases were calculated, and their mechanical stability was determined.

For the considered and predicted phases, we have presented their optimized crystallographic data, electronic structure, elastic moduli, and hardness. It was established that the band structure of all the considered low-temperature model phases of silver sulfide has a band gap, which indicates their semiconductor properties.

The predicted alternative low-temperature phases of silver sulfide (Ag_2_S), especially nanocrystalline ones, can also have many possible applications. One can expect that these phases will be suitable for biological and medical applications as biomarkers, and the predicted low-temperature Ag_2_S phases can be used for the manufacture of such electronic devices as photovoltaic cells, photoconductors, and infrared radiation detectors.

## Figures and Tables

**Figure 1 nanomaterials-13-02638-f001:**
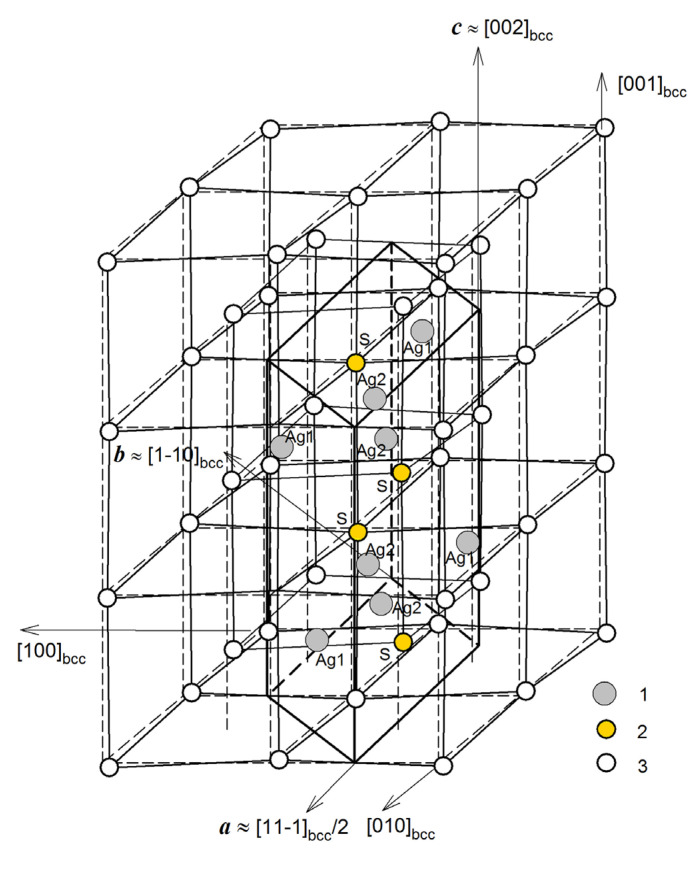
Position of the monoclinic (space group *P*2_1_/*c*) unit cell of *α*-Ag_2_S (acanthite) in the distorted bcc nonmetal sublattice of argentite. (○) and (*●*) are S atoms located beyond and inside (within) a monoclinic unit cell of α-Ag_2_S (acanthite), respectively; (*●*) are Ag atoms in a monoclinic unit cell of *α*-Ag_2_S.

**Figure 2 nanomaterials-13-02638-f002:**
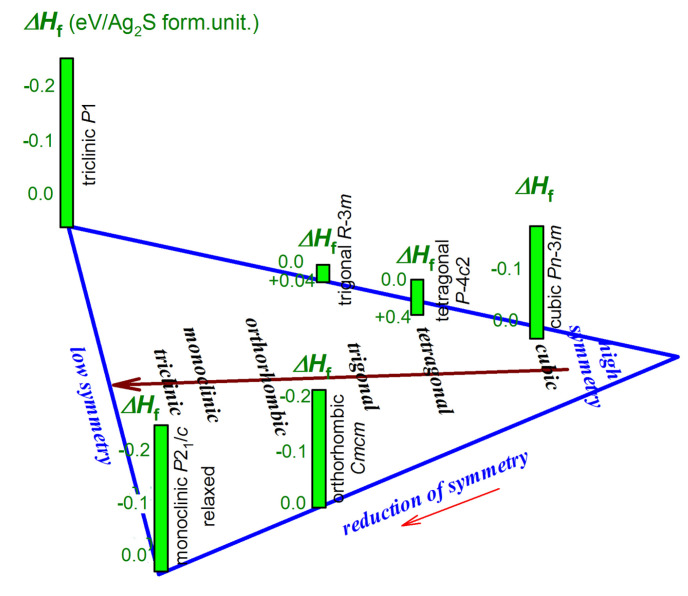
Variation of the formation enthalpies *H*_f_ of some the predicted Ag_2_S phases with symmetry reduction.

**Figure 3 nanomaterials-13-02638-f003:**
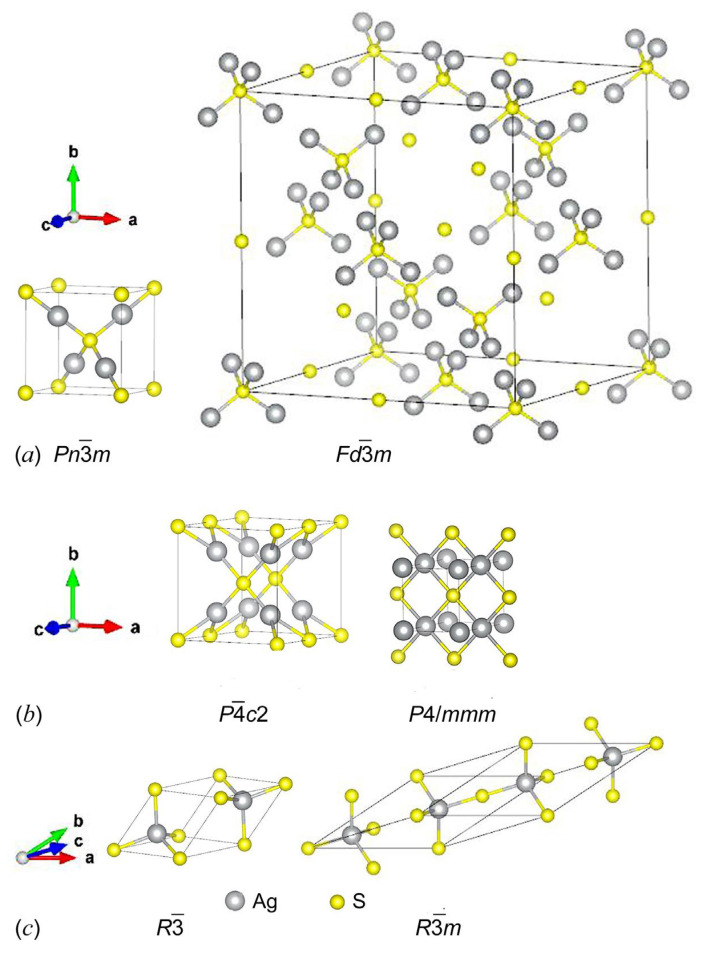
Unit cells of possible ordered Ag_2_S structures derived from argentite: (**a**) model cubic Ag_2_S phases with space groups No.224-$Pn\bar{3}m$ and No.227-$Fd\bar{3}m$, (**b**) model tetragonal Ag_2_S phases with space groups No.116-$P\bar{4}c2$ and No.123-*P*4/*mmm*, and (**c**) model trigonal Ag_2_S phases with space groups No.148-$R\bar{3}$ and No.166-$R\bar{3}m$.

**Figure 4 nanomaterials-13-02638-f004:**
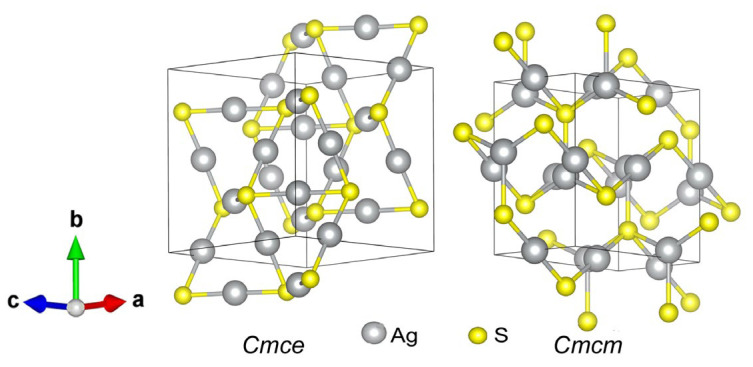
Unit cells of model orthorhombic Ag_2_S phases with space groups No.64-*Cmce* and No.63-*Cmcm*.

**Figure 5 nanomaterials-13-02638-f005:**
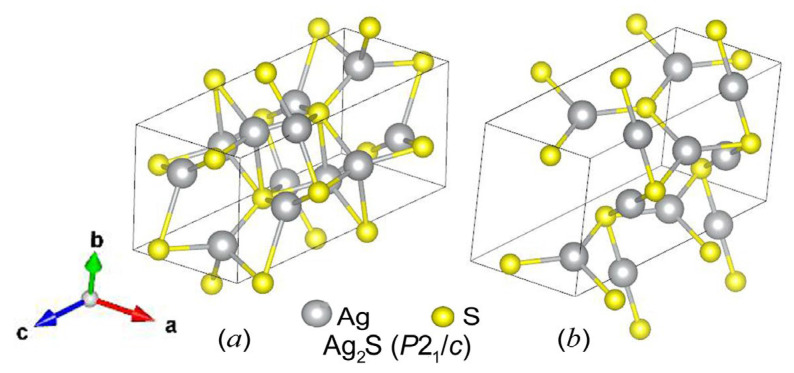
Unit cells of monoclinic (space group *P*2_1_/c) phases with *α*-Ag_2_S (acanthite) structure: (**a**) structure before relaxation according to diffraction experiment data [46], (**b**) structure after relaxation.

**Figure 6 nanomaterials-13-02638-f006:**
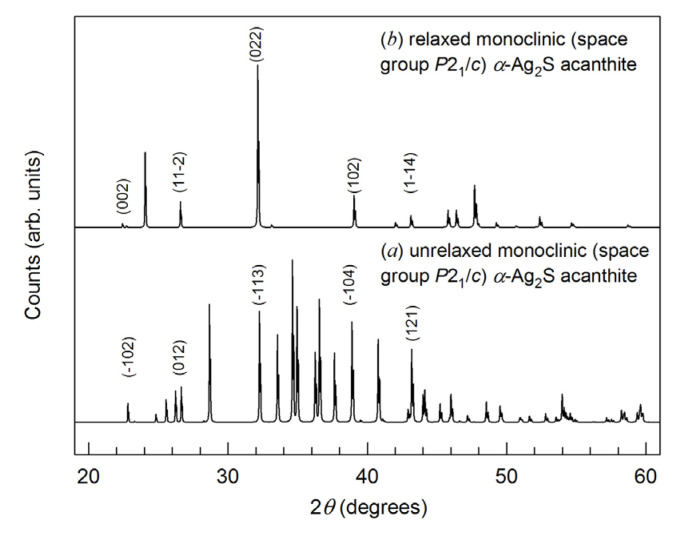
Calculated XRD patterns (Cu*Kα*_1,2_ radiation) of monoclinic (space group *P*2_1_/*c*) phases with an *α*-Ag_2_S (acanthite) structure: (**a**) unrelaxed structure, (**b**) relaxed structure. The theoretical XRD pattern of unrelaxed acanthite coincides with high accuracy with the experimental XRD pattern of coarse-crystalline monoclinic (space group *P*2_1_/*c*) *α*-Ag_2_S (acanthite) [65].

**Figure 7 nanomaterials-13-02638-f007:**
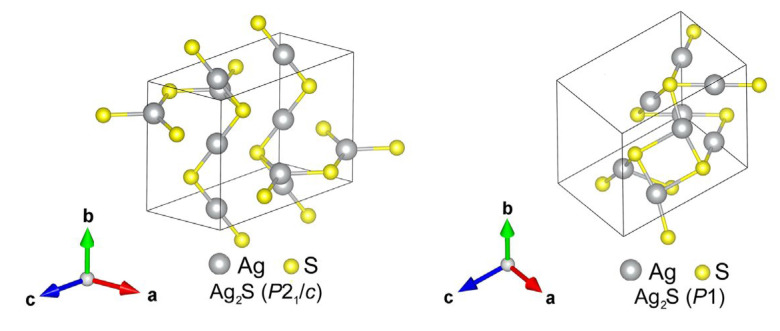
Unit cell of the model monoclinic (space group *P*2_1_/*c*) Ag_2_S phase (**left**), with the lowest formation enthalpy Δ*H*_f_ = −0.219 eV/form.unit among the predicted monoclinic structures, and unit cell of the model triclinic (space group *P*1) Ag_2_S phase (**right**) with the lowest formation enthalpy Δ*H*_f_ = −0.223 eV/form.unit.

**Figure 8 nanomaterials-13-02638-f008:**
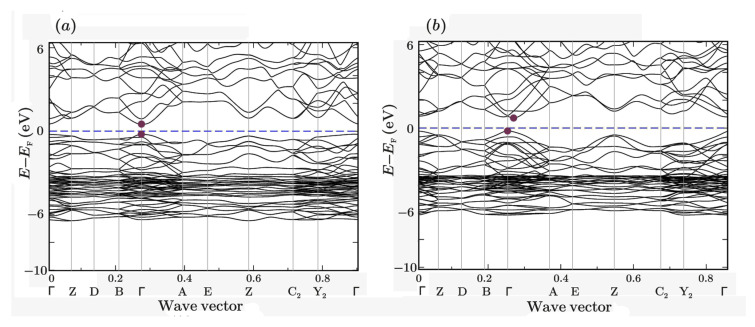
Band structure of (**a**) unrelaxed and (**b**) relaxed monoclinic (space group *P*2_1_/*c*) *α*-Ag_2_S (acanthite).

**Figure 9 nanomaterials-13-02638-f009:**
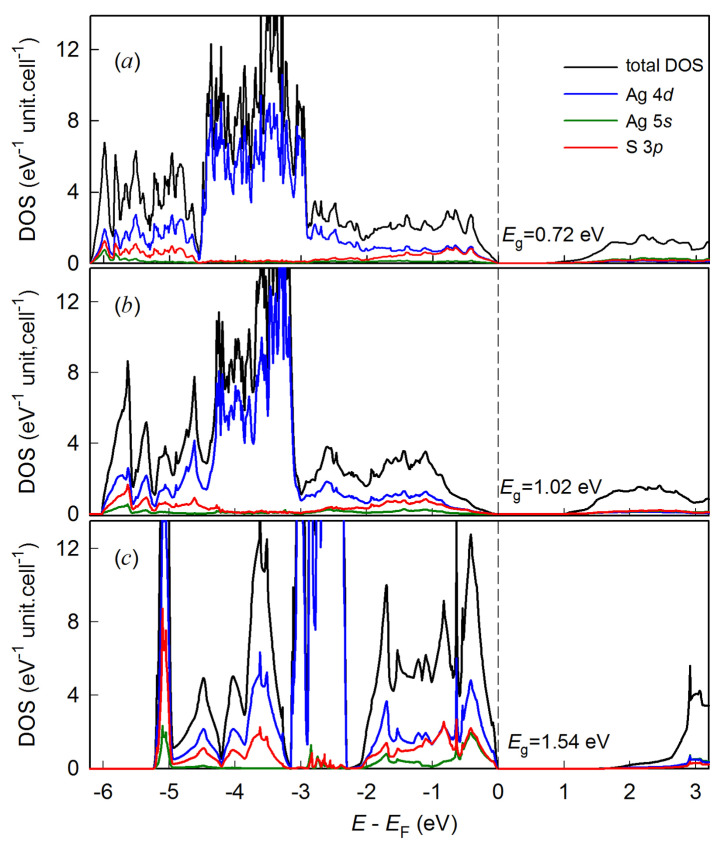
Total and partial Ag 4*d-*, Ag *5s*-, and S 3*p*-densities of electronic states (DOS) computed for (**a**) unrelaxed and (**b**) relaxed monoclinic (space group *P*2_1_/*c*) *α*-Ag_2_S (acanthite) and (**c**) the proposed ordered cubic (space group $Pn\bar{3}m$) Ag_2_S phase, which is a derivative of the argentite structure.

**Table 1 nanomaterials-13-02638-t001:** Calculated cohesion energy (*E*_coh_), formation enthalpy (Δ*H*_f_), rotational (*N*_rot_), translational (*N*_tr_), and total (*N*_tot_) reductions of symmetry for predicted silver sulfide phases.

Symmetry and Space Group	Number of Atoms		^1^ *Z*	^2^ *E*_phase_, eV	* E * _coh_	^3^ Δ*H*_f_	* N * _rot_	^4^ *N*_tr_	* N * _tot_
	* N * _Ag_	* N * _S_			eV/(Ag_2_S form.unit)				
Cubic (No.224-$Pn\bar{3}m$)	4	2	2	−19.498	−8.272	−0.191	1	1.5	1.5
Cubic (No.227-$Fd\bar{3}m$)	32	16	16	−95.749	−4.507	3.573	1	8	8
Tetragonal (No.116-$P\bar{4}c2$)	8	4	4	−36.655	−7.687	0.394	6	2	12
Tetragonal (No.123-*P*4/*mmm*)	2	1	1	−8.2445	−6.767	1.313	3	0.5	1.5
Trigonal (No.148 -$R\bar{3}$)	2	1	1	−9.516	−8.039	0.042	8	0.5	4
Trigonal (No.166 -$R\bar{3}m$)	4	2	2	−19.033	−8.039	0.041	4	1	4
Orthorhombic (No.64-*Cmce*)	16	8	8	−78.216	−8.300	−0.219	6	5	30
Orthorhombic (No.63-*Cmcm*)	8	4	4	−39.027	−8.280	−0.199	6	2	12
Orthorhombic (No.36-*Cmc*2_1_)	8	4	4	−39.027	−8.280	−0.199	12	2	24
Orthorhombic (No.19-*P*2_1_2_1_2_1_)	8	4	4	−38.384	−8.119	−0.038	12	2	24
Monoclinic (No.11-*P*2_1_/*m*)	4	2	2	−19.514	−8.280	−0.199	12	1	12
Monoclinic (No.4-*P*2_1_)	4	2	2	−19.532	−8.289	−0.208	24	1.5	36
Monoclinic (No.14-*P*2_1_/*c*) unrelaxed acanthite	8	4	4	−38.361	−8.113	−0.033	12	2	24
Monoclinic (No.14-*P*2_1_/*c*) relaxed acanthite	8	4	4	−39.028	−8.280	−0.199	12	2.5	30
Monoclinic (No.14-*P*2_1_/*c*)	8	4	4	−39.107	−8.300	−0.219	12	3	36
Triclinic (No.1-*P*1)	8	4	4	−39.125	−8.304	−0.223	48	2	96

^1^ The number of Ag_2_S formula units in the predicted silver sulfide phase (*Z* coincides with the number of S atoms in the predicted phase); ^2^ *E*_phase_: the energy of the predicted Ag_2_S structure obtained via the DFT calculation using the VASP code; ^3^ the estimated relative standard deviations of the formation enthalpies do not exceed 0.1%; ^4^ the reduction in translational symmetry is estimated as relative change in the unit cell volume of the predicted phase in comparison with the unit cell volume of argentite.

**Table 2 nanomaterials-13-02638-t002:** Model predicted orthorhombic Ag_2_S structures.

Space Group	Atom	Position and Multiplicity	Atomic Coordinates in the Model Structures		
			*x/a* ≡ *x/a*_orthorh_	*y/b* ≡ *y/b*_orthorh_	*z/c* ≡ *z/c*_orthorh_
* No.63-*Cmcm*	Ag1	4(*b*)	0	0.5	0
	Ag2	4(*c*)	0	0.04657	0.25
	S	4(*c*)	0	0.68496	0.25
** No. 64-*Cmce*	Ag1	8(*e*)	0.25	0.22162	0.25
	Ag2	8(*f*)	0	0.02841	0.75496
	S	8(*f*)	0	0.72452	0.39559
*** No.36-*Cmc*21	Ag	8(*b*)	0	0.04705	0.06242
	S	4(*a*)	0	0.31480	0.56242
**** No.19-*P*212121	Ag1	4(*a*)	0.00985	0.77359	0.64205
	Ag2	4(*a*)	0.15219	0.38829	0.45552
	S	4(*a*)	0.12297	0.00356	0.34389

* Parameters of the predicted unit cell (space group *Cmcm*): *a* = 0.461442 nm, *b* = 0.738306 nm, *c* = 0.791054 nm, *V* = 0.269500 nm^3^, and *Z* = 4; ** parameters of the predicted unit cell (space group *Cmce*): *a* = 0.832398 nm, *b* = 0.834841 nm, *c* = 0.862424 nm, *V* = 0.599315 nm^3^, and *Z* = 8; *** parameters of the predicted unit cell (space group *Cmc*21): *a* = 0.461617 nm, *b* = 0.737986 nm, *c* = 0.790727 nm, *V* = 0.269374 nm^3^, and *Z* = 4; **** parameters of the predicted unit cell (space group *P*212121): *a* = 0.438446 nm, *b* = 0.694766 nm, *c* = 0.733716 nm, *V* = 0.223502 nm^3^, and *Z* = 4.

**Table 3 nanomaterials-13-02638-t003:** Monoclinic (space group No.14-*P*2_1_/*c*) *α*-Ag_2_S (acanthite) structures before relaxation [29] and after relaxation.

Monoclinic (Space Group *P*2_1_/*c*) *α*-Ag_2_S	Atom	Position and Multiplicity	Atomic Coordinates in the Model Structures		
			*x/a* ≡ *x/a*_mon_	*y/b* ≡ *y/b*_mon_	*z/c* ≡ *z/c*_mon_
* Unrelaxed unit cell [46]	Ag1	4(*e*)	0.07157	0.48487	0.80943
	Ag2	4(*e*)	0.27353	0.67586	0.56247
	S	4(*e*)	0.4922	0.2341	0.13217
** Relaxed unit cell	Ag1	4(*e*)	0.04498	0.74996	0.47750
	Ag2	4(*e*)	0.50004	0.00001	0.25002
	S	4(*e*)	0.31581	0.25002	0.34210

* Parameters of the unrelaxed unit cell (space group *P*21/*c*): *a* = 0.42264 nm, *b* = 0.69282 nm, *c* = 0.953171 nm, *α* = 90°, *β* = 125.554°, *γ* = 90°, *V* = 0.227068 nm^3^, and *Z* = 4; ** parameters of the relaxed unit cell (space group *P*21/*c*): *a* = 0.435628 nm, *b* = 0.791975, *c* = 0.871257 nm, *α* = 90°, *β* = 116.111°, *γ* = 90°, *V* = 0.269911 nm^3^, and *Z* = 4.

**Table 4 nanomaterials-13-02638-t004:** Model predicted monoclinic (space group No.14-*P*2_1_/*c*) Ag_2_S structure.

Atom	Position and Multiplicity	Atomic Coordinates in the Model Structures		
		*x/a* ≡ *x/a*_mon_	*y/b* ≡ *y/b*_mon_	*z/c* ≡ *z/c*_mon_
Ag1	4(*e*)	0.52641	0.24248	0.47333
Ag2	4(*e*)	0.02658	0.25188	0.47346
S	4(*e*)	0.52441	0.39461	0.22983

Parameters of the unrelaxed unit cell (space group *P*21/c*): a* = 0.590253 nm, *b* = 0.863974 nm, *c* = 0.836796 nm, *α* = 90°, *β* = 135.1413°, *γ**=* 90°, *V =* 0.301002437 nm^3^, and *Z =* 4.

**Table 5 nanomaterials-13-02638-t005:** Model triclinic (space group No.1–*P*1) Ag_2_S structure.

Atom	Position and Multiplicity	Atomic Coordinates in the Model Structures		
		*x/a* ≡ *x/a*_tricl_	*y/b* ≡ *y/b*_tricl_	*z/c* ≡ *z/c*_tricl_
Ag1	1(*a*)	0.86160	0.11768	0.65392
Ag2	1(*a*)	0.18251	0.45397	0.34767
Ag3	1(*a*)	0.68702	0.45957	0.35685
Ag4	1(*a*)	0.02431	0.56922	0.00840
Ag5	1(*a*)	0.52371	0.56411	0.00802
Ag6	1(*a*)	0.36410	0.10713	0.66484
Ag7	1(*a*)	0.00327	0.00177	0.00153
Ag8	1(*a*)	0.50720	0.00312	0.01042
S1	1(*a*)	0.62789	0.41309	0.70515
S2	1(*a*)	0.90766	0.15612	0.30911
S3	1(*a*)	0.11466	0.85369	0.70273
S4	1(*a*)	0.42086	0.71554	0.31158

Parameters of the triclinic unit cell (space group *P*1): *a* = 0.593602 nm, *b* = 0.706832 nm, *c* = 0.781798 nm, *α* = 116.4408°, *β* = 110.9258°, *γ* = 91.3246°. *V* = 0.267753731 nm^3^, and *Z* = 4.

**Table 6 nanomaterials-13-02638-t006:** Calculated bulk (*B*_H_) and shear (*G*_H_) elastic moduli (GPa), inverse Pugn’s ratio, *k* = *G*_H_/*B*_H_, Poisson’s ratio (*μ*), Young’s modulus (*E*_H_), and Vickers hardness (*H*_V_) (GPa) for polycrystalline predicted Ag_2_S (silver sulfides) with the lowest formation enthalpies.

Symmetry, Space Group	* G * _V_	* G * _R_	* G * _H_	* B * _V_	* B * _R_	* B * _H_	* k *	* μ *	* E * _H_	* H * _V_	
										Equation (5)	Equation (6)
Orthorhombic (*Cmce*)	13.0	3.6	8.3	18.0	10.3	14.1	0.588	0.277	20.0	2.0	1.0
Orthorhombic (*Cmcm*)	11.8	8.6	10.2	34.9	21.9	28.4	0.360	0.359	25.9	1.2	1.5
Monoclinic (*P*2_1_)	13.4	−2.3	5.6	13.7	−2.9	5.4	1.032	0.133	12.2	3.2	1.0
Monoclinic (*P*2_1_/*c*) unrelaxed acanthite	11.6	8.6	10.1	34.3	27.9	31.1	0.326	0.371	25.9	1.0	1.5
Monoclinic (*P*2_1_/*c*) relaxed acanthite	11.3	8.3	9.8	35.1	21.5	28.3	0.348	0.363	25.0	1.1	1.4
Monoclinic (*P*2_1_/*c*)	−0.8	12.4	5.8	25.3	61.3	43.3	0.134	0.445	15.5	0.2	0.9

## Data Availability

The data presented in this study are available on request from the corresponding author.

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
