# Peer review of "Low-Temperature Predicted Structures of Ag2S (Silver Sulfide)"

_nanomaterials, 2023, doi:10.3390/nano13192638_

Round 1

Reviewer 1 Report

Manuscript ID: nanomaterials-2595278; Title: “Low-Temperature Predicted Structures of Ag2S Silver Sulfide” In this paper the author focused on alternative structural models for low-temperature modifications of silver sulfide Ag2S in addition to the known monoclinic structure of acanthite phase including the nanocrystalline state of alternative phases.
The novelty of the study is missing. A detailed comparison is needed, and the current state of the study is not enough for publication. I did not have a good impression of this manuscript. The author didn't explain/provide their results. So, in my opinion, I recommend publication only after major revisions.
If the author is interested in revising the manuscript, clarify the following questions.
1)  The abstract of this paper does not look effective; the author needs to rewrite the abstract and include the motivation and necessity of this research. Needs to explain the importance of this research.
2) The author needs to include an introduction, what are the Advantages and Disadvantages, and the literature survey is not sufficient to prove the objective of this work.
3) Page No:1; The author has written equations; I think the author needs to represent equations with numbers for better understanding.
4) The author needs to include the appropriate working principal figure in the introduction for better understanding.
5) In the result and discussion, I didn't find any comparison study with the results in this paper. The author needs to include a clear comparison of previous results Vs present results.
6) The Conclusions of this paper are too long; the author needs to shorten them.

The language of the manuscript should be checked in detail. It is far from academic writing language. Get help from professionals or natives if necessary.

Author Response

Answers to the 1st reviewer’s remarks:

1) First remark: The author needs to rewrite the abstract and include the motivation and necessity of this research. Needs to explain the importance of this research.

Answer on first remark:

According to this remark of reviewer, the abstract is rewritten.  

2) Second remark: The author needs to include in an introduction, what are the Advantages and Disadvantages.

Answer on second remark:

In accordance with remark 2, section “Introduction” is supplemented with new references [9], [11], [13-21], as well as an explanation of the advantages of the research performed on the prediction of previously unknown low-temperature phases of silver sulfide (see page 3).

3) Third remark:  The authors have written equations; I think the authors needs to represent equations with numbers for better understanding

Answer on third remark:

Examples of the equations (2) and (3) are presented with numbers on page 5 for the better understanding.

Clarification that added on page 5:

“For example, cubic (space group ) Ag2S structure is characterized by following quantities Ephase = -19.498 eV, NAg = 4, NS = 2, and Z = 2 (see Table 1), and EAg-cond = -10.863 eV, NAg-cond = 4, ES-cond = -528.168 eV, NS-cond = 128, EAg-at = -0.198 eV, ES-at = -1.081 eV (see Table S1, Supplementary Materials). According to eq.(2), the formation enthalpy DHf of cubic (space group ) Ag2S structure for these quantities is  

DHf = [-19.498 - (-10.863×4/4) - (-528.168×2/128)]/4 = [-19.498+10.863+8.252]/4 = -0.382/2 = -0.191 eV.

According to eq.(3), the cohesion energy Ecoh  of cubic (space group ) Ag2S structure for these quantities is equal

Ecoh = [-19.498 - (-0.198×4) - (-1.081×2)]/2 = [-19.498+0.792+2.162]/2 = -16.544/2 = -8.272 eV.

It is these values ofDHf and Ecoh that are given in Table 1 for cubic (space group ) Ag2S structure.”

4) Fourth remark:  The author needs to include the appropriate working principal figure in the introduction for better understanding.

Answer on fourth remark:

A new plot of the formation enthalpies DHf for some predicted Ag2S phases with different symmetries is included in the “Introduction” section (see pages 3, 4).

5) Fifth remark: The author needs to include a clear comparison of previous results Vs present results.

Answer on 5th remark:

Comparisons of the obtained results with data from previous works are added to Sections 3.2 (pages 12, 13) and 3.4 (pages 15 and 17).     

Text that added on page 12:

“In general, calculations of model structures of silver sulfide Ag2S using the USPEX code [31-33] showed that a reduction of the symmetry of Ag2S phases from cubic, tetragonal and trigonal to orthorhombic, monoclinic and, especially, triclinic crystallographic systems is accompanied by a decrease in their formation enthalpy ΔHf and the emergence of the most energetically favorable structures. This was not previously known. Indeed, the transformation of high-temperature cubic (space group ) b-Ag2S argentite into any model Ag2S structure will occur with a decrease in symmetry.”     

6) Sixth remark: Conclusions of this paper are too long; the author needs to shorten them.

Answer on 6th remark:

In accordance with remark 6, we have rewritten the conclusions in a more concise form.

The authors should like to convey our heartfelt thanks to the 1st reviewer for interest in our work and for his useful remarks.

Reviewer 2 Report

The manuscript deals with several theoretical studies of different phases of silver sulfide.

The article is carefully written and appears correctly supported by bibliography, although several references are very old. Furthermore, there is a rather high percentage of self. citation 8-9 of 47, which should be avoided as much as  possible.

In my opinion some other minor problems should be reviewed:

In the paraghaph L659-667 use subsript for formulae.

In ref 2 the family name of the first author is missing.

Author Response

Answers to the 2nd reviewer’s comments and remarks:

1) First comment: There is a rather high percentage of self citation, which should be avoided.

Answer on first comment:

The authors own only 12 references out of 62 references presented in the Reference List.

2) Second remark: In the paragraph L659-667 use subsript for formulae.

Answer on second remark:

The indicated errors have been corrected.

3) Third remark: In ref 2 the family name of the first author is missing.

Answer on third remark:

The family names of all authors are indicated in all references, including reference [2].

The authors thank the referee for his remarks and comment.

Reviewer 3 Report

The manuscript by S.I. Sadovnikov et al. presents computational study of possible silver sulfide polymorphs. This study reports the output of the evolutionary code and simply lists the obtained results. Therefore, the manuscript is suitable for a more specialized journal of computational chemistry. However, to fit the scope of the Nanomaterials journal, the authors should perform major improvement of the whole text.

1) The Introduction is focused solely on the polymorphs of silver sulfide and transitions between them. This technical information is more like a part of Discussion. At the same time the Introduction should explain why these polymorphs are so interesting for the readers and provide a brief overview of recent developments, where the silver sulfide materials and nanomaterials are comprehensively described.

2) The manuscript and Supporting Info are not complete. They don’t show any crystallographic details for the phases with orthorhombic symmetry or lower. These data should be included, because it is crucial to understand the conclusions of this work. All parameters, including unit cell and atomic positions should be given with the estimated standard deviations.

3) The manuscript shows low scientific soundness due to simple listing of the results. All computational results should be treated with a pinch of salt: the authors should analyze all chemical aspects of the predicted phases, including bond lengths and coordination environment and compare those with the known silver chalcogenides and related compounds. Without this analysis the study looks like a reading of the “out” file.

4) “Literature theoretical data on the structure and lattice parameters of some model Ag2S phases with triclinic, monoclinic and orthorhombic symmetry are presented in the form of a database on the websites of such the Materials Projects as mp-32284, mp-556225, mp-1095694, mp-32669, mp-36216, mp-31053, mp-32791, mp-610517, and others.” Please, provide the proper reference and http link to these data.

5) Was a geometry optimization been used in VASP calculations? If Yes, provide the details of calculations of forces.

Author Response

Answers to the 3rd reviewer’s remarks:

1) First remark: The Introduction should explain why these polymorphs are so interesting for the readers.

Answer on first remark:

In accordance with the first remark of the reviewer, the “Introduction” section has been supplemented with new references [24-30] and a brief explanation of interest in the discussed low-temperature Ag2S polymorphs (see page 3).

Explanation that added on page 3:

“Interest in the discussed low-temperature Ag2S polymorphs is due to the possibility of their wide application. All forms of silver sulfide Ag2S have attracted much attention [2, 7, 24]. Monoclinic silver sulfide a-Ag2S is a semiconductor at temperatures <~420-450 K, and body-centered cubic sulfide b-Ag2S exhibits superionic conductivity at temperatures greater than 452 K. Nanocrystalline silver sulfide is a versatile semiconductor for use in various optoelectronic devices, such as photocells and photoconductors, infrared sensors [2, 25, 26]. The use of nanocrystalline silver sulfide is promising for the creation of Ag2S/Ag heteronanostructures intended for use in memory devices and resistance switches. Their action is based on the transformation of a-Ag2S acanthite into b-Ag2S argentite and the formation of a conducting channel between silver Ag and superionic b-Ag2S argentite [27-30]. The band gap of the predicted low-temperature Ag2S phases may be different from that of acanthite, which will expand the potential applications of silver sulfide.”     

2) Second remark: Crystallographic details for the phases with orthorhombic symmetry should be included. All parameters, including unit cell and atomic positions should be given with the estimated standard deviations.

Answer on second remark:

Crystallographic data for two orthorhombic phases (space groups Cmc21 and P212121) are additionally included in Table 2 (see page 8). Parameters of the unit cells and atomic positions of all the predicted Ag2S phases, presented in tables 2-5, are the calculated but no experimental values and consequently have no standard deviations.

3) Third remark: The authors should analyze all chemical aspects of the predicted phases, including bond lengths and coordination environment.

Answer on the remark 3:

The Ag-S bond lengths for all the predicted Ag2S structures are given in Table S5 (see Supplementary Material). The coordination numbers for the nearest environment of Ag atoms in the predicted Ag2S phases with different symmetries range from 2 to 3.

The Ag-S bond length is 0.232-0.252 nm depending on the symmetry of the predicted Ag2S structure (see page 12).

4) Fourth remark: Please, provide the proper reference and http link to these data (mp-32284, mp-556225, mp-1095694, mp-32669, mp-36216, mp-31053, mp-32791, mp-610517).

Answer on fourth remark:

References [13-20] on listed projects are given in the beginning of section 3.1 (see page 5).

5) Fifth remark: Was a geometry optimization been used in VASP calculations? If Yes, provide the details of calculations of forces.

Answer on fifth remark:

A geometry optimization was used in VASP calculations.  We optimized the unit cell parameters and atomic positions using identical settings for the DFT calculations of their energies. Structural optimization was stopped if the forces acting on the atoms were less than 10-3 eV×Å-1. This procedure is described in the second (see page 4) and penultimate (see page 5) paragraphs of Section 2:

“To find equilibrium unit cell parameters and atomic positions, we performed stepwise relaxation [39] using VASP implemented conjugate-gradient and quasi-Newton RMM-DIIS algorithms [40].” and “…we made the optimization of the unit cell parameters and positions of atoms using the identical settings of the DFT calculations of their energies. The structural optimization stopped if forces acting on atoms were less than 10-3 eV×Å-1. The forces were calculated according Hellmann-Feynmann theorem.”

The authors thank the referee for his constructive useful remarks and comments.

Round 2

Reviewer 1 Report

Thank you to the author for addressing all the raised comments from reviewers. The authors have carried out the modifications and the corrections to the queries raised and indeed have provided more information in the revised version. Therefore, I recommend this manuscript for publication in Nanomaterials as it is without further modifications.

 Minor editing of English language required

Author Response

Answer to the 1st reviewer

Thank you very much for your positive decision.

Reviewer 3 Report

The authors didn't follow the given recomendations, the manuscript should be major revised.

1) It is not sufficient to add one paragraph to the Introduction and made it convinient for the readers. Currently, the Introduction is a technical text. It should be rewritten.

2) The standard deviations of computational quantities are estimated using two values of the limiting parameter. The authors should perform two sets of calculations using 1x10-3 eV A-3 and 5x10-2 eV A-3 for residual forces and extract the standard deviations. It is a common practice.

3) The authors have completely ignored the explanation of chemical properties of the calculated compounds (coordination environment,bond lengths). Without this discussion the manuscript is not ready for publication.

Author Response

Answers to remarks of a repeated report of the 3rd reviewer:

1) First remark: Currently, the Introduction is a technical text. It should be rewritten..

Answer on first remark:

In accordance with this remark, the “Introduction” section has been rewritten and the technical details of the introduction have been shortened.

Two new sentences added to the “Introduction” section on page 2:

“The unusual physical and structural properties of the high temperature phases, namely the enhanced ionic conductivity, liquid-like behavior of silver sublattice [8, 9], the uncertainty in the positions of silver atoms [10], and the alleged ordering have always raised questions.”

“This relationship between the structures of a-Ag2S and b-Ag2S also raises the questions regarding the existence of other ordered phases based on high-temperature b- Ag2S with the disordered silver sublattice.”

New sentence added to the “Introduction” section on page 3:

“These authors considered three hypothetical models of structural ordering for the high temperature argentite phase, in which a cooperative ionic transport is possible. These models were only used for electronic structure calculations, and the evaluation of their thermodynamic stability have not been provided.”

New paragraph added to the “Introduction” section on page 3:

“The Open Quantum Materials Database (OQMD) [16] and Materials Project databases [17] contain many other hypothetical structures of Ag2S which, apparently, were simulated on the basis of structural similarity using silver chalcogenides or related systems. Both resources consider the monoclinic (space group P21) structure proposed by Kashida et al [15] as the best low-temperature model for silver sulfide rather than the experimental acanthite phase. Thus, the real number of structures related exactly to Ag2S as well as the anticipated sequences of phase transitions remain unknown.”

New text added to the “Introduction” section on pages 3, 4:

“Indeed, silver sulfide Ag2S, along with lead, zinc, copper, cadmium and mercury sulfides, belongs to the most in-demand semiconductors. This is an excellent material for fabrication of heterostructures [35] which can also be used in solar cells [36], IR detectors [29, 37], resistive switches and nonvolatile memory devices [31-34]. Besides, Ag2S is promising for solar energy conversion into electricity [38]. The application of semiconducting nanostructured silver sulfide in biology and medicine as biosensor is based on the quantum size effects that influence the optical properties of the material. Semiconducting Ag2S/noble metal heterostructures are treated as candidate materials for application in photocatalysis [39]. In particular, this is due to the narrow band gap of silver sulfide (about 0.9 eV).

Main fields of application of silver sulfide include microelectronics, biosensorics and catalysis.

High-performance atomic resistive switches are a promising type of nonvolatile memory devices; here, read/write operations are implemented involving ion exchange [40]. In the case of cation migration, silver and silver sulfide heteronanostructures are most widely used [31-34]. The Ag2S/Ag heteronanostructures combine the sulfide (semiconductor or ionic conductor depending on the structure) and silver (electronic conductor) [41].

Design of fluorescent labels (biolabels, biomarkers) based on Ag2S quantum dots for applications in biology and medicine seems to be quite promising [42-45]. Nanocrystalline silver sulfide and heteronanostructures based on this substance are treated as effective materials for catalysis and photocatalysis [39, 46, 47]. Nanocrystalline silver sulfide has also been considered as an effective thermoelectric material [48]. Silver sulfide is of interest as thermoelectric material owing to the reversible transition between the monoclinic semiconducting (a-Ag2S) and cubic superionic (b-Ag2S) phases. This allows the thermoelectric effect in Ag2S to be realized near the phase transition temperature with silver sulfide acting as thermoelement. 

Thus, silver sulfide Ag2S has a lot of possible applications. The prediction of new low-temperature phases of silver sulfide Ag2S will allow to expand its potential use considerably.”

2) Second remark: The authors should perform two sets of calculations using 1´10-3 eV A-3 and 5´10-2 eV A-3 for residual forces and extract the standard deviations.

Answer on second remark:

According to second remark, standard deviations of the formation enthalpies have been calculated. The estimated relative standard deviation of the formation enthalpies of Ag2S phases is given in the footnote to Table 1.

Additional explanatory text on page 13 of the manuscript:

“We assessed the influence of residual forces on the standard deviation of the formation enthalpies of the considered Ag2S phases. In particular, when assessing the impact of the residual forces on the unrelaxed acanthite structure, we found that the absolute deviations in formation enthalpy did not exceed 0.00034 eV/(Ag2S form.unit) or 0.00085 eV×Å-1×atom-1 in case of changing the force threshold from 0.002 eV×Å-1 (to 0.001 eV×Å-1. The relative deviation of the formation enthalpy does not exceed 0.1%.  For smaller values of the limiting parameter, it was impossible to complete the relaxation of the structure due to computational limitations. The other considered phases were more susceptible to relaxation, so deviations of their energies are very likely to be within this 0.1% range. This amount of uncertainty is enough to rank the Ag2S structures by their stability. For instance, the difference in the formation enthalpies, taking into account their uncertainties, for the two best structures (triclinic and orthorhombic with space groups P1 and Cmce) is about 2 %. Thus, the deviations in the formation enthalpies of structures with different symmetries are very small, which makes it possible to distinguish between these structures.”

3) Third remark: The authors have completely ignored the explanation of chemical properties of the calculated compounds (coordination environment, bond lengths).

Answer on the remark 3:   

The authors provided information on the Ag-S bond lengths in the predicted Ag2S structures in the first revised version of the manuscript (see Table S5, Supplementary Material) and also noted that the coordination numbers for the nearest environment of silver atoms with sulfur atoms depend on the symmetry of the predicted Ag2S phases. In this recorrected version of the manuscript, information on the coordination environment of Ag atoms and Ag-S bond lengths is presented in more detail.

New text added on pages 13 and 14 (end of subsection 3.1):

“In high-temperature body-centered cubic b-Ag2S argentite at a temperature close to 453 K, the length of bonds Ag1-S and Ag2-S are 0.24307 and 0.25691nm, respectively. The coordination numbers of Ag1 (b positions) and Ag2 (j positions) atoms are 6 and 1, respectively. In the crystal lattice of a-Ag2S acanthite, the length of bonds Ag1-S and Ag2-S are 0.25113 and 0.25475 nm, respectively (see Table S5, Supplementary Material). The coordination numbers of Ag1 and Ag2 atoms are the same and equal to 4. Thus, taking into account the temperature change in the lattice parameters of silver sulfide during the transition from argentite to acanthite, the Ag1-S and Ag2-S bond lengths increased, and the coordination numbers of Ag atoms became the same.  The coordination numbers of silver atoms in the predicted Ag2S structures are also less than 6. The smallest coordination number of Ag atoms, equal to 1, is observed for Ag-S1 bonds with a length of 0.23883 nm of the cubic (space group ) Ag2S phase. Coordination numbers of Ag atoms equal to 2 are observed for Ag-S bonds with lengths of 0.23556 nm and 0.24306 nm in the cubic (space group ) and orthorhombic (space group Cmce) Ag2S phases, as well as for all Ag-S bonds in a predicted triclinic (space group P1) Ag2S phase. The coordination number of Ag atoms in Ag-S bonds of tetragonal (space group ) and trigonal (space groups  and ) silver sulfides is 3. The coordination number of Ag atoms equal to 3 is also observed in a relaxed monoclinic (space group P21/c) acanthite. Ag atoms have a fourfold coordination environment by S atoms in unrelaxed monoclinic (space group P21/c) acanthite and tetragonal (space group P4/mmm) silver sulfide.

In the structures having the lowest formation enthalpies (triclinic and orthorhombic with space groups P1 and Cmce), the Ag-S bond lengths range from 0.2431 to 0.2496 nm. The Ag-S bond lengths of the relaxed structure of acanthite are somewhat outside these limits and demonstrate an increasing disproportion. One of the bonds has the length of 0.2524 nm, while another 0.2403 nm. This disproportion is considerably larger in the another proposed structures. The biggest difference in bond lengths is reached for the tetragonal (space group ) and one of the cubic (space group ) models. Therefore, from the chemical point of view, the formation of such structures seems unreasonable. The cubic (space group ) structure after relaxation contains chemically isolated sulfur atoms. The nearest distance to the silver atom is 0.6545 nm. Presumably, this peculiarity makes the cubic (space group ) structure the most unfavorable from all the considered structural models of silver sulfide. Another suggested cubic (space group ) structure has no disproportion in bond lengths. Its nearest Ag-S distance (0.2356 nm) is slightly less than that of the best structures. Nevertheless, the formation energy of the cubic (space group ) ordered structure is close to that of acanthite. Therefore, from all the suggested cubic, tetragonal and trigonal models, only the cubic (space group ) structure can be considered as a candidate for new phase of silver sulfide.”

The authors thank the referee for his constructive useful remarks.

Round 3

Reviewer 3 Report

The manuscript is ready for publication in the present form.